# Endothelial Injury Syndromes after Allogeneic Hematopoietic Stem Cell Transplantation: Angiopetin-2 as a Novel Predictor of the Outcome and the Role of Functional Autoantibodies against Angiotensin II Type 1 and Endothelin A Receptor

**DOI:** 10.3390/ijms24086960

**Published:** 2023-04-09

**Authors:** Dionysios Vythoulkas, Ioanna Lazana, Christos Kroupis, Eleni Gavriilaki, Ioannis Konstantellos, Zoi Bousiou, Spiros Chondropoulos, Marianna Griniezaki, Anna Vardi, Konstantinos Gkirkas, Aggeliki Karagiannidou, Ioannis Batsis, Maria Stamouli, Ioanna Sakellari, Panagiotis Tsirigotis

**Affiliations:** 1Hematology Division, 2nd Department of Internal Medicine, Propaedeutic, “ATTIKON” University Hospital, National and Kapodistrian University of Athens, 12462 Athens, Greece; 2Clinical Biochemistry and Molecular Diagnostics, “ATTIKON” University Hospital, National and Kapodistrian University of Athens, 12462 Athens, Greece; 3Hematology and Bone Marrow Transplantation Department, “G. Papanikolaou” General Hospital, 57010 Thessaloniki, Greece

**Keywords:** graft-versus-host disease, transplant-associated thrombotic microangiopathy, endothelial injury syndromes, angiotensin II type 1 receptor, endothelin A receptor, angiopoetin-2

## Abstract

Transplant-associated thrombotic microangiopathy (TMA) occurs in a significant percentage of patients after allogeneic stem cell transplantation (allo-SCT) and is associated with significant morbidity and mortality. The aim of the present study was to examine the association of serum angiopoetin-2 (Ang2) levels and the presence of antibodies against angiotensin II type 1 (AT1R) and ndothelin A Recreptor (ETAR) with the outcome of patients with TMA and/or graft-versus-host disease (GVHD) after allo-SCT. Analysis of our data showed that elevated serum Ang2 levels at the time of TMA diagnosis are significantly associated with increased non-relapse mortality and decreased overall survival. To our knowledge, this is the first study demonstrating an association between raised Ang2 levels and poor outcomes in patients with TMA. Antibodies against AT1R (AT1R-Abs) and ETAR (ETAR-Abs) were detected in 27% and 23% of the patients, respectively, but there was no association between the presence of autoantibodies and the outcome of patients with TMA. However, a significant finding was the strong positive correlation between the presence of AT1R-Abs with the occurrence of chronic fibrotic GVHD, such as scleroderma and cryptogenic organizing pneumonia, raising the possibility of the contribution of autoantibodies in the pathogenesis of fibrotic GVHD manifestations.

## 1. Introduction

Angiopoietin 2 (Ang2) plays a key role in the pathogenesis of endothelial cell damage in many different inflammatory diseases [1]. Ang1 and Ang2 are peptides that act as ligands for the same tyrosine kinase receptor Tie2, expressed on the surface of endothelial cells (EC). Binding of Ang1 to its receptor induces an anti-inflammatory response through downregulation of surface adhesion molecules and promotion of EC survival. On the contrary, Ang2 binding to the same receptor results in the opposing effects by inducing EC apoptosis and secretion of inflammatory cytokines. Previous studies in the setting of allogeneic stem cell transplantation have shown that Ang2 is a major factor contributing to the pathogenesis of systemic endothelial injury [2].

Increased serum levels of Ang2 at the time of transplant have been correlated with the development and the severity of endothelial cell injury-related syndromes, such as sinusoidal obstruction syndrome (SOS), transplant-associated microangiopathy (TA-TMA), idiopathic pneumonia syndrome (IPS), and diffuse alveolar hemorrhage (DAH) [2,3]. Previous studies have not shown any association between serum Ang2 levels and the subsequent development of acute graft-versus host disease (GVHD). However, increased levels of Ang2 were significantly associated with increased GVHD-associated mortality [4].

Angiotensin II type 1 receptor (AT1R) and endothelin A receptor (ETAR) are G protein coupled receptors (GPCRs) widely expressed in many cell types, but mainly on the surface of vascular endothelial and smooth muscle cells [5]. Binding of angiotensin II to AT1R induces vasoconstriction, secretion of aldosterone and sodium retention, resulting in blood pressure regulation and maintenance of homeostasis [6]. Similarly, ETAR activation after binding to endothelin induces vascular smooth muscle contraction and plays a key role in the regulation of normal blood flow [7]. However, these receptors have pleotropic effects, and through their activation contribute to the regulation of collagen synthesis, proliferation and cell migration, which are important processes for tissue repair and wound healing [8].

Previous studies have shown the presence of Abs against AT1R (AT1R-Abs) and ETAR (ETAR-Abs) in solid organ transplant recipients with chronic allograft dysfunction, thus raising significant concerns for the direct pathogenetic effect of Abs in the process of chronic rejection [9,10,11,12].

AT1R-Abs and ETAR-Abs do not activate complement or mediate opsonization and are therefore not harmful to receptor expressing cells. AT1R-Abs and ETAR-Abs are characterized as “functional” antibodies, because they induce activation of their target receptor in a similar manner as their natural ligands [13].

Moreover, antibody binding to AT1R and ETAR induces a sustained and significantly prolonged activation of the target receptor, as compared to the activation induced by the natural ligands [14]. The prolonged activation is probably due to the skewing of receptor binding. The natural ligands activate both AT1R and ETAR, but also the receptors AT2R (angiotensin type 2 receptor) and ETBR (endothelin B receptor), which help in the counterbalance of the main receptors by inducing the opposite effects [15]. On the contrary, AT1R-Abs and ETAR-Abs activate only the main receptors AT1R and ETAR, resulting in skewing of receptor activation and in disruption of normal homeostasis [16].

Regarding the pathophysiology of chronic allograft rejection, AT1R-Abs and ETAR-Abs are considered not only biomarkers but also potential contributors to allograft dysfunction [9,13]. However, the exact mechanisms as well as the severity of allograft injury mediated by AT1R-Abs and ETAR-Abs are not clearly understood. Moreover, allograft dysfunction is not observed in all patients with positive AT1R-Abs and ETAR-Abs, and antibody titers associated with the development of injury vary widely among different studies [12,17,18].

Possible mechanisms directly contributing to poor transplant outcome in patients with autoantibodies are briefly mentioned below: (1) Enhanced signaling driven by the binding of autoantibodies to its receptor results in vasoconstriction, ischemia and severe organ microvasculature dysfunction. (2) The presence of AT1R-Abs and ETAR-Abs have also been correlated with development of fibrosis in liver allografts [19]. Similarly, in kidney transplant recipients, high AT1R mRNA expression was associated with renal interstitial fibrosis [20]. For both liver and kidney transplant recipients, development of fibrosis is significantly associated with poor long-term graft survival [21]. (3) Immune effectors such as cytotoxic T-cells express AT1R, and activation of this receptor after binding of autoantibody induces the migration of T cells to areas of injury and infection [22], while activation of ETAR through ETAR-Abs results in the recruitment of neutrophils [23]. Recruitment of immune effectors to the transplanted organ results in a vicious cycle of further immune activation and inflammation.

The role of AT1R-Abs and ETAR-Abs in the pathogenesis of endothelial injury syndromes after allogenic stem cell transplantation (allo-SCT) has not been previously studied. Therefore, in the present study we examined the association of Ang2 serum levels with the outcome of 52 patients with TMA and/or GVHD after allo-SCT. We also tested for the presence of antibodies against AT1R and ETAR, and we examined the association of antibodies with the outcome of TMA and GVHD.

## 2. Results

### 2.1. Patient Characteristics

We studied 52 patients who underwent an allo-HSCT from 2016 to 2019 at two BMT Units in Greece and who were diagnosed with GVHD and/or TA-TMA. Demographic information and pretransplant data were uniformly collected on all patients. For analysis, the patients were divided into two cohorts, depending on the presence or absence of TMA, as follows: (i) Cohort 1 consisted of 20 consecutive patients with samples at the time of TMA diagnosis, (ii) Cohort 2 included 32 consecutive patients with moderate-severe acute and/or chronic GVHD, treated during the same time period, and had available stored serum samples.

The two cohorts were well balanced for basic transplant and disease characteristics, such as, sex, age, disease, type of donor, graft source, and conditioning regimen, as illustrated in Table 1. More specifically, Cohort 1 consisted of 13 males and 7 females with a median age at the time of transplant of 49 years (range, 35–65). Diagnosis of TMA was established in a median of 80 days after graft infusion (range, 45–740). TMA developed in the context of concurrent acute and chronic GVHD in 14 and 2 patients, respectively, while in 4 patients TMA was diagnosed in the absence of GVHD. Cyclosporine was dose reduced or discontinued in all patients at the time of TMA diagnosis. Plasma exchange and/or plasma infusion was administered in 8 and 5 patients, respectively, while 7 patients received treatment with eculizumab. TMA resolved in 10 out of 20 patients, while 10 patients had refractory disease and died from causes directly associated with TMA. One patient with acute myeloid leukemia (AML) died due to leukemia relapse, while another patient with Myelodysplastic syndrome (MDS) in complete remission died from infection. Complete remission was achieved in 4 out of 7 patients treated with eculizumab, and the drug was successfully discontinued in all cases.

Cohort 2 consisted of 16 males and 16 females with a median age at the time of transplant of 53 years (range, 27–68). Severe acute GVHD grade III-IV and severe chronic GVHD developed in 8 and 13 patients, respectively. TMA did not develop in any of the patients in Cohort 2 during their post-transplant course.

### 2.2. Serum Ang2 Levels and Autoantibodies against AT1R and ETAR

There was no difference in serum Ang2 levels between patients with TMA (median 86 ng/mL, 95% CI, 48–150) and patients with GVHD (median 73 ng/mL, 95% CI, 48–85) (Cohort 1 vs. 2, *p* = 0.2) at the time of TA-TMA or GVHD diagnosis.

However, in the whole patient population (Cohort 1 and 2), the serum Ang2 levels were significantly higher in the group of patients who died from transplant-related complications during the follow-up period. More specifically, the median serum Ang2 level was 91.6 ng/mL (95% CI, 73–152) in patients who died versus 56.1 ng/mL (95% CI, 43–81) in the group of patients who remained alive during the follow-up period, (*p* = 0.01), (Figure 1).

In total, 14 out of 52 (27%) patients had AT1R-Abs titers above 10 IU/mL, while 12 out of 52 (23%) had ETAR-Abs titers above 10 IU/mL and therefore were considered positive for the presence of AT1R-Abs and ETAR-Abs, respectively. High-titer AT1R-Abs and ETAR-Abs (defined as titers above 17 IU/mL) were detected in 5 (9.6%) and 3 (5.7%) out of 52 patients, respectively. In Cohort 1 (TMA), 6 out of 20 (30%) patients were positive for the presence of both AT1R-Abs and ETAR-Abs. In Cohort 2 (GVHD), 8 out of 32 (25%) and 6 out of 32 (19%) patients were positive for the presence of AT1R-Abs and ETAR-Abs, respectively. There was no statistical difference in the percentages of AT1R-Abs and ETAR-Abs positive patients between Cohorts 1 and 2.

Spearman’s rank analysis showed a statistically significant positive correlation between AT1R-Abs and ETAR-Abs titers (rho = 0.919, 95% CI, 0.862–0.953), *p* < 0.001, (Figure 2a).

Patients with higher serum Ang2 levels had higher AT1R-Abs titers. Spearman’s rank analysis revealed a significant correlation between AT1R-Abs titers and serum Ang2 levels (rho = 0.318, 95% CI, 0.048–0.543), *p* = 0.023, (Figure 2b).

### 2.3. Non-Relapse Mortality—All Patients

The cumulative incidence of non-relapse mortality (NRM) for the whole group of patients at 5 years was 56% (95% CI, 26–78%). In multivariate analysis, the only parameters associated with NRM were serum levels of Ang2 and diagnosis of TMA in the context of GVHD (Table 2). Patients with GVHD and TMA had a significantly higher probability of NRM (59%, 95% CI, 29–74%), as compared with patients with GVHD in the absence of TMA (31%, 95% CI, 15–48%), *p* = 0.006, (Figure 3a). Patients with Ang2 serum levels above the median had a significantly higher probability of NRM (52%, 95% CI, 31–70%), as compared with those with Ang2 serum levels below the median (17%, 95% CI, 5–34%), *p* = 0.01, (Figure 3b).

### 2.4. Outcome of Patients with TMA

For patients with TMA, the median overall survival was 32 months, while the cumulative incidence of NRM and OS at 5-years was 54% (95% CI, 29–74%) and 36% (95% CI, 11–61%), respectively. In multivariate analysis, the only parameter associated with NRM was the serum levels of Ang2 (Table 2). The cumulative incidence of NRM in the group of patients with Ang2 serum levels above the median (75%, 95% CI, 33–93%) was significantly increased in comparison with the cumulative incidence of NRM in patients with Ang2 serum levels below the median (33%, 95% CI, 7–63%), *p* = 0.04, (Figure 4a). Patients with Ang2 serum levels below the median had a significant trend for higher OS as compared with the group of patients with Ang2 serum levels above the median (Figure 4b). EASIX score was not statistically associated with NRM in the group of patients with TMA.

### 2.5. Autoantibodies against AT1R and GVHD-Manifestations

In total, 48 patients suffered from GVHD (16 patients from Cohort 1 and 32 patients from Cohort 2). Twelve out of 48 patients had chronic GVHD associated with fibrotic manifestations (9 patients with scleroderma and 3 patients with cryptogenic organizing pneumonia). Eleven out of 12 patients with chronic fibrotic GVHD tested positive for the presence of AT1R-Abs, while among the group of 36 patients without fibrotic GVHD, only 3 tested positive for AT1R-Abs (*p* < 0.001). In more detail, 3 out of 3 patients with cryptogenic organizing pneumonia (COP), 8 out of 9 patients with sclerodermatous chronic-GVHD, and only 3 out of 36 patients without scleroderma and/or COP were positive for the presence of AT1R-Abs (Figure 5a).

Similar findings were observed when the analysis was performed only for patients with high-titer AT1R-Abs positivity. More specifically, 5 out of 12 (42%) patients with chronic fibrotic GVHD were positive for the presence of high-titer AT1R-Abs, while only 1 out of 36 (1.8%) without fibrotic GVHD was positive for high-titer AT1R-Abs (*p* < 0.001).

Receiver operating characteristic (ROC) curve analysis showed that AT1R-Abs titers above 9.2 IU/mL had 89% sensitivity and 92% specificity for the diagnosis of either chronic sclerodermatous GVHD or COP after allo-SCT (AUC = 0.932, 95% CI, 0.822–0.933, *p* < 0.001), (Figure 5b).

## 3. Discussion

Emerging evidence supports the role of endothelial dysfunction in the pathogenesis of non-infectious transplant-related complications, contributing to the increased morbidity and mortality documented after allo-SCT [24]. On this basis, a number of studies have investigated the significance and contribution of various angiogenic factors, such as the vascular endothelial growth factor (VEGF) and Ang2, in the development of such complications, along with their correlation with patient outcomes. More specifically, Luft et al., [25] described a high VEGF/Ang2 ratio in patients with steroid-refractory GvHD, compared to those with sensitive disease, whereas Porkholm et al., reported a correlation with the incidence of intestinal and skin/liver GvHD with the pre- and post-transplant Ang2 levels, respectively [26]. Ueda et al., reported a strong prognostic value of serum Ang2 levels for transplant-related complications with endothelial cell damage and leukemia relapse [3].

In this study, we examined Ang2 serum levels, in patients with TMA (cohort 1) and those with acute or chronic GvHD (cohort 2). We found no significant difference between the two at the time of diagnosis. This may be attributed to the fact that the majority of patients in Cohort 1 had co-existing GvHD, which is known to be associated with high Ang2 levels [27]. It is, however, of interest that higher Ang2 levels were found to be correlated with shorter OS, which is in keeping with previous reports, documenting a significantly lower 5-year OS in patients with high Ang-2 levels [26]. Furthermore, in multivariate analysis, Ang2 levels and the co-current diagnosis of TMA were found to be the only parameters predicting high NRM, which is again consistent with previous findings [26].

Interestingly, multivariate analysis in the group of patients with TMA showed that the only parameter associated with increased NRM was increase serum Ang2 levels at the time of diagnosis. To our knowledge, this is the first study showing that serum Ang2 levels may be used as a predictive biomarker for the outcome of patients with TMA.

The contribution of non-HLA autoantibodies, and particularly of AT1R- and ETAR-Abs, in the development of vascular inflammation and graft rejection after solid organ transplantation has been well established [9,28,29,30]. However, there is no information on the impact of such antibodies in the context of allo-SCT, apart from a single study of 3 allo-SCT patients with refractory hypertension (3/3) and diarrhoea (2/3) who were found to have elevated AT1R-Abs [31]. Interestingly, the introduction of an angiotensin receptor blocker in these patients resulted in symptom resolution, suggesting a potential pathogenetic role for AT1R-Abs.

This is the first study, to our knowledge, investigating the presence of AT1R-Abs and ETAR-Abs in the allo-SCT setting and exploring the potential association of these antibodies with the development and outcome of TMA and/or GvHD. Our results have shown that 9.6% and 5.7% of patients had high titres (>17 units/mL) of AT1R-Abs and ETAR-Abs, respectively. It is of note that different cut-off levels have been used by different studies [31,32,33,34,35], making it difficult to interpret and compare results from different studies. However, the cut-off value of >10 units/mL has been adopted by the majority of studies [31,32,33,34], and it is the same that we used in our study as well, whereas the cut-off of >17 units/mL was used to define ‘high-titer’ patients.

We also detected a positive correlation between AT1R-Abs and ETAR-Abs and AT1R-Abs and Ang2 levels. A similar association between ETAR-Abs and AT1R-Abs has been described in a pediatric kidney transplant cohort, prompting monitoring of such antibodies, in addition to HLA donor-specific antibodies, to allow for targeted therapeutics [34].

Significantly higher percentage of patients with fibrotic chronic GvHD were found to have high-titer levels of AT1R-Abs, as compared to those without fibrotic GvHD (42% vs. 2.5%, *p* = 0.001). This may be explained by the fact that AT1R has been implicated in the proliferation of fibroblasts and collagen production and in accelerated inflammatory cell infiltration [36]. Furthermore, AT1R activation has been implicated in the pathogenesis of lung and liver fibrosis, as well as in the fibrosis of chronic-GvHD in a murine chronic-GvHD model [37,38]. Raised AT1R-Abs and ETAR-Abs have also been associated with micro-vasculopathy and allograft loss in heart, lung, liver and multivisceral transplantation [34].

More interestingly, we demonstrated that AT1R-Abs have a positive diagnostic capacity with high sensitivity and specificity for the development of sclerodermatous GvHD and/or COP after allo-SCT. This, again, is the first time, to our knowledge, that such an association has been described, prompting further investigation.

Despite the very promising results, our study carries significant limitations, such as: (i) the retrospective nature of our data, (ii) the limited sample size, restricting the inclusion of additional factors in multivariable analyses, (iii) the absence of a control cohort group (such as healthy individuals or allo-HSCT patients with no TMA or GvHD) and (iv) the lack of serial sample collection at different time points during the disease course.

In summary, to our knowledge, this is the first study demonstrating an association between raised Ang2 levels and poor outcomes (shorter OS and higher NRM) in patients with TMA. More interestingly, raised AT1R-Abs levels were found to be positively correlated with the occurrence of fibrotic chronic GvHD after allo-HSCT, suggesting that testing for these antibodies may have clinical utility and especially in the era of multiple angiotensin and endothelin A receptor blockade agents. However, larger, prospective studies are required to validate these results prior to their clinical translation.

## 4. Materials and Methods

### 4.1. Patients and Study Design

The study was approved by IRB and bioethics committee of ATTIKO University Hospital (IRB: 10992). Patients gave written informed consent before entry into the study.

Diagnosis of TMA was based on the criteria proposed by the International Working Group for TMA definition criteria (IWG) [39], whereas GVHD was diagnosed and graded according to established NIH criteria [40,41]. Serum samples were collected from all patients with TMA and/or GVHD at the time of diagnosis and stored at −80 °C.

Treatment of TMA included discontinuation of immunosuppression, plasma exchange and/or plasma infusion, or administration of eculizumab. The criteria used for evaluation of response to treatment have been described previously [42]. EASIX score at the time of diagnosis was estimated for patients with TMA with the use of the following formula: [(LDH x Creatine)/PLT] [43].

### 4.2. Ang2 and Antibodies against AT1R and ETAR

Analysis of the Ang2 levels was performed by enzyme-linked immunosorbent-based assay (ELISA), according to the instructions of the manufacturer (ABCAM PLC, Cambridge, UK). AT1R-Abs and ETAR-Abs quantitation were determined by ELISA (CellTrend GmbH, Luckenwalde, Germany). Sera were diluted 1:100 and tested in duplicates. AT1R-Ab and ETAR-Ab concentrations were determined by a standard curve. The cutoff of ≥10 units/mL was used as ‘positive’ for the presence of autoantibodies [31,32,33], whereas patients with levels >17 units/mL were considered as ‘positive with high-titers’. Serum and plasma samples from patients collected at the time of diagnosis of TMA or GVHD were stored at the biobanking facilities of the two hospitals.

### 4.3. Statistical Analysis

Statistical analysis aimed to identify pre-treatment factors associated with NRM in patients with TMA and/or GVHD after allo-SCT, and to estimate the proportion of patients tested positive for the presence of autoantibodies against AT1R and ETAR. Overall survival (OS) was defined as the time from transplantation to last follow-up or death due to any cause, while NRM was defined as the time from transplantation to last follow-up, or death due to any cause in patients free of disease. Categorical variables between groups were compared using the two-sided Fisher’s exact test. Competing risk analysis was used for estimating the cumulative incidence of NRM. Gray’s test was used for univariate analysis of GVHD and NRM, while Fine and Gray proportional hazards regression model was used for multivariate analysis. The following variables entered in the multivariate analysis model: (1) Age at the time of allo-SCT (above vs. below the median), (2) Sex (male vs. female), (3) Presence of TMA (yes vs. no), (4) Ang2 serum levels (above vs. below the median), (5) Presence of Abs against AT1R and ETAR (yes vs. no), (6) EASIX score (above vs. below the median). Receiver operating characteristic (ROC) curve analysis was performed in order to identify the predictive accuracy of anti-AT1R-Abs titers for the diagnosis of specific GVHD manifestations. Statistical analysis was performed with the use of easy R and Medcalc statistical software [44].

## Figures and Tables

**Figure 1 ijms-24-06960-f001:**
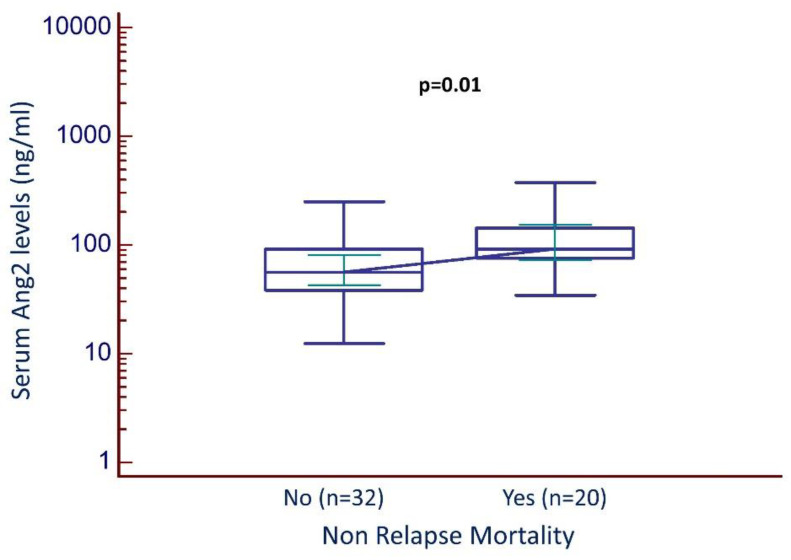
Serum Ang2 levels and non-relapse mortality (all patients).

**Figure 2 ijms-24-06960-f002:**
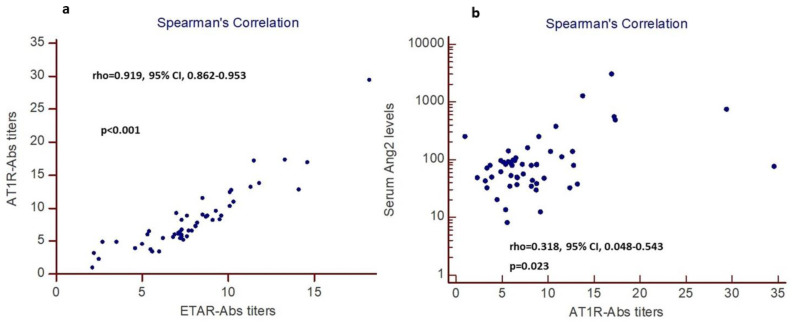
Spearman’ rank analysis (n = 52 patients): (**a**) positive correlation between AT1R-Abs and ETAR-Abs titers (patients with high AT1R-Abs have also high ETAR-Abs titers), (**b**) positive correlation between AT1R-Abs titers and serum Ang2 levels (patients with high AT1R-Abs titers also have high serum Ang2 levels).

**Figure 3 ijms-24-06960-f003:**
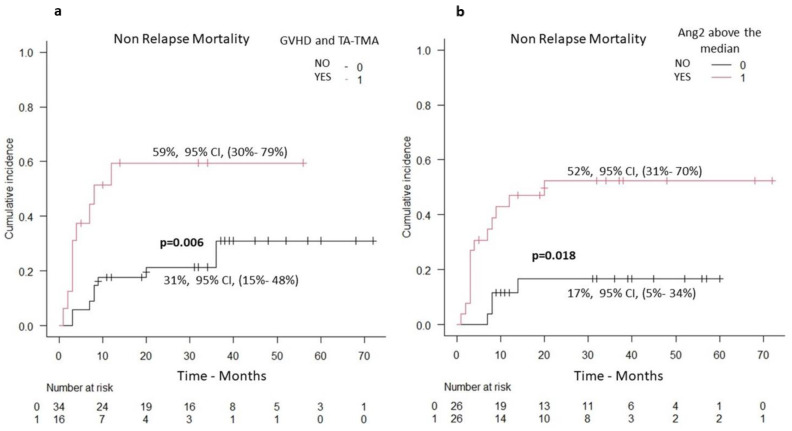
Non-relapse mortality (n = 52 patients): (**a**) patients with TMA and concurrent GVHD versus patients with GVHD in the absence of TMA (increased NRM in the group of patients with TMA and concurrent GVHD), (**b**) patients with serum Ang2 levels above versus below the median (increased NRM in the group of patients with serum Ang2 levels above the median).

**Figure 4 ijms-24-06960-f004:**
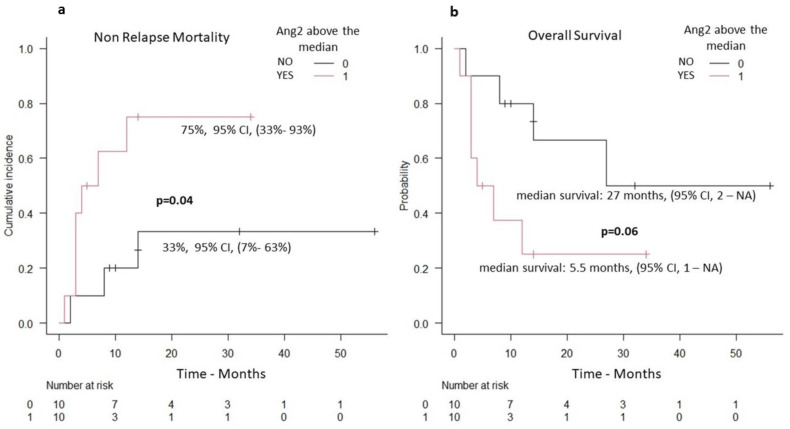
Patients with TMA and serum Ang2 levels (n = 20 patients): (**a**) non-relapse mortality (increased NRM in the group of patients with serum Ang2 above the median), (**b**) overall survival (decreased OS in the group of patients with serum Ang2 above the median).

**Figure 5 ijms-24-06960-f005:**
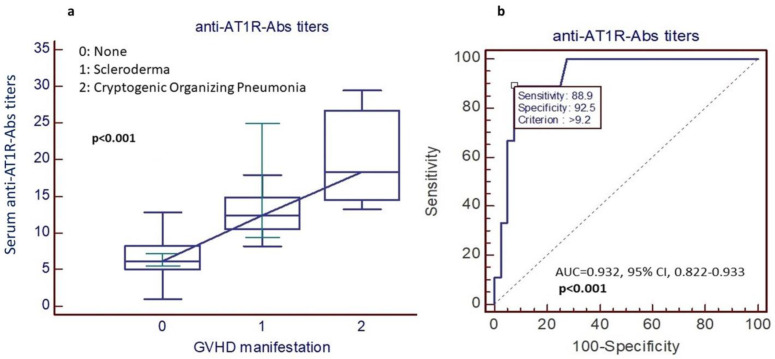
(**a**) AT1R-Abs and chronic GVHD with fibrotic manifestations (9 patients with scleroderma and 3 patients with COP have higher AT1R-Abs titers in comparison with patients without fibrotic GVHD manifestations), (**b**) the predictive value of AT1R-Abs titers.

**Table 1 ijms-24-06960-t001:** Patient characteristics.

Cohorts (1, 2) *	TMA (No = 20 pts)	GVHD w/o TMA (No = 32 pts)	Statistics
**Sex** (Male/Female)	13/7	16/16	*p* = n.s
**Age**, median (range)	49.5 years, (35–65)	53.5 years, (27–68)	*p* = n.s
**Disease (%)** AML MDS ALL MPN CLL	10/20 3/20 5/20 2/20	17/32 3/32 6/32 5/32 1/32	*p* = n.s
**Donor** MUD MMUD MRD	11/20 5/20 4/20	12/32 8/32 12/32	*p* = n.s
**Graft source** PBSC	20/20	32/32	*p* = n.s
**Conditioning** MAC RIC	14/20 6/20	21/32 11/32	*p* = n.s
**GVHD** Moderate Chronic Severe Chronic Acute (grade-II) Severe Acute (III-IV)	1/20 1/20 9/20 5/20	11/32 13/32 12/32 8/32	*p* = n.s
**Outcome **** Alive in CR Dead Cause of Death Relapse TMA NRM ^&^	8/20 12/20 1/20 10/20 1/20	18/32 14/32 5/32 NA 9/32	*p* = n.s

* TMA: Cohort 1 consisted of 20 patients with TMA with or without GVHD, Cohort 2 consisted of 32 patients with GVHD without TMA, ** at the time of last follow-up, AML: acute myeloid leukemia, MDS: myelodysplastic syndrome, ALL: acute lymphoblastic leukemia, MPN: myeloproliferative neoplasm, CLL: chronic lymphocytic leukemia, MUD: matched unrelated donor, MMUD: mismatched unrelated donor, MRD: matched related donor, PBSC: peripheral blood stem cells, MAC: myeloablative conditioning, RIC: reduced intensity conditioning, **^&^** NRM: non-relapse mortality (except TMA).

**Table 2 ijms-24-06960-t002:** Multivariate analysis for Non-Relapse Mortality.

Parameter	Hazard Ratio	95% CI	Significance
All patients (Cohort 1 and 2)			
Serum Ang2 levels (above vs. below the median	3.298	1.168–9.307	*p* = 0.024
TA-TMA and GVHD vs. GVHD in the absence of TA-TMA	2.916	1.133–7.506	*p* = 0.026
Only patients with TA-TMA (Cohort 1)			
Serum Ang2 levels (above vs. below the median	7.859	1.296–47.650	*p* = 0.025

## Data Availability

The datasets generated during and/or analysed during the current study are available from the corresponding author on reasonable request.

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
