# Peer review of "Endothelial Injury Syndromes after Allogeneic Hematopoietic Stem Cell Transplantation: Angiopetin-2 as a Novel Predictor of the Outcome and the Role of Functional Autoantibodies against Angiotensin II Type 1 and Endothelin A Receptor"

_ijms, 2023, doi:10.3390/ijms24086960_

Round 1

Reviewer 1 Report

The study addresses an important new topic and appears to be well conducted with interesting results but there are some issues that need to be taken into account.

 * Authors must insert line numbers; it was difficult to make a comment without it.

 * Introduction:

   It would be interesting to add more scientific background on Ang2 and its patho-physiological role in the development of GVHS and TMA. You should also expand the links with prior publications in the area.

   More details on the role played by AT1R-abs and ETAR-abs in graft rejection.

   References need to be updated.

   The paragraph beginning "In addition to chronic allograft dysfunction....." is out of scope, please delete.

 * Materials and Methods: The patient's consent and the study approval should be written under the title of "Patients and study design".

 * Results:

   Table-1: The number of patients divided into acute or chronic GVHD does not correspond to the number written in the paragraph beginning with "The two cohorts were well balanced for basic transplant.... Please clarify.

   Table-1: The same for the Outcome. Please provide more details to understand.

   Page 4: The paragraph beginning "Spearman rank analysis...." the 3rd line: what do you mean by "All 12 patients positive for...."? Do you mean in the 2 cohort studies? Please clarify.

   Please correct "anti-AT1R-Abs" into AT1R-Abs.

   Page 7: The title "Autoantibodies against AT1R and GVHD-manifestations" the 1st paragraph: "Eleven out of 12 patients with chronic GVHD...." The number does not correspond to the number written in table-1. Please clarify.

 * Please use constant way throughout the manuscript to write the abbreviation "graft versus-host disease" as either GVHD or GvHD.

Author Response

Our response to reviewer's comments is attached as word doc

Reviewer 2 Report

Vythoulkas et al. performed an interesting clinical study that involved patients with TMA and/or graft-versus-host disease (GVHD) after Allo-SCT. This study showed a clear relation between elevated angiopoietin II levels and poor outcomes in patients with TMA and no relation between ATI1R and ET1R antibodies in clinical outcomes at the testing time. Finally, AT1R antibody levels showed a correlation with chronic fibrotic GVHD. The authors discuss the results and mark the limitations of this study. However, the authors must address the following comments:

Overall comment: Very interesting study.

As a recommendation: In the title of the study is not mentioned Ang2, but one of the relevant findings of this study is the correlation between Ang2 and patient outcome. Highlighting Ang2 could be more appealing to the reader.

Detailed comments:

In the introduction section (2nd paragraph, last line), should be “…important processes.”

Figure 1: Please add Y axis label. Moreover, please provide more information about the figure (i.e., white, and red squares) in the figure legend section.

Figure 2: Please add information in figure legend in order to improve the understanding of the figure.

Figure 3: Please add information in figure legend in order to improve the understanding of the figure.

Figure 4: Please add information in figure legend in order to improve the understanding of the figure.

Figure 5: Figure 5a, please add Y axis label. Moreover, please provide more information about the figure (i.e., red squares and S.E.M green bar) in the figure legend section.

Author Response

(The authors gave the same response as above.)

Reviewer 3 Report

The aim of this study was to examine the relationship between angipoietin-2 (Angpt2) and antibodies specific for AT1 and ET-A receptors with complications of allogenic hematopoietic stem cell transplantation including thrombotic microangiopathy and graft vs. host diseases. The study was performed in 52 patients divided into 2 cohorts, one with GVHD without TMA and the other with TMA (most of them with coexisting GVHD). The main results are that Anpt2 concentration was positively correlated with non-relapse mortality whereas both AT1R-Ab and ETAR-Ab were not. However, AT1R-Ab were positively correlated with fibrotic GVHD. Angpt2 level and the percentage of antibody-positive patients was similar in both cohorts. The titer of AT1R-Ab was correlated with the titer of ATAR-Ab.

The study is of interest, however, there are also important concerns to be addressed.

1)     Introduction: it is stated that the relationship with “development and outcome of TMA and GVHD” was examined. However, GVHD was present in almost all patients, so the relationship with the development of GVHD could not be established.

2)     Abbreviations “TMA” and “TAM” are used interchangeably in the manuscript. This nomenclature should be unified.

3)     Was there any difference in the results between patients with and without GVHD in the TMA cohort? Because there were only 4 patients without GVHD in this cohort, authors could consider to restrict the analysis to those with GVHD in this group.

4)     The nomenclature of the cohorts in table 1 is not consistent with the text; according to the text cohort 1 includes those with TMA regardless of whether GVHD was present or not and cohort 2 those with GVHD without TMA.

5)     Section 2.3 vs. 2.4: outcome of all pooled patients and outcomes of patients with TMA are analyzed in separate sections. What about outcomes in TMA- cohort?

6)     Statistical analysis: how was the risk of NRM analysed? What statistical method was used?

7)     Were all consecutive patients fulfilling the inclusion criteria included or were they selected from the bigger group? Was the required sample size quantified before the study?

Author Response

(The authors gave the same response as above.)

Reviewer 4 Report

The authors provide an interesting study examining the correlation between serum angiopoietin-2 levels and serum antibody titres targeting angiotensin II type 1 and endothelin A receptor in the clinical outcomes for patients presenting with transplant associated thrombotic microangiopathy and/or graft-vs-host disease. In examining the patient history and samples of a total population of 52, the authors determine that an elevated serum level of Ang2 at the time of transplant associated thrombotic microangiopathy correlates with a decreased chance of survival. Taken together, this was an interesting and novel examination of an area of research requiring more investigation.

In reviewing the manuscript, I had a number of small observations. The following should be considered by the authors when preparing a suitable revision.

1.       It would be useful to clarify how old the serum/biological samples were at the time of analyses.

2.       Can the n-number be clearly stated in all graphs and tables to clarify any and all inclusions and exclusions from the data sets?

3.       The authors should revise their graphs/figures an ensure that aspects such as labelling with what is being measured, the units, etc. are clear and obvious.    

Author Response

(The authors gave the same response as above.)

Round 2

Reviewer 3 Report

The manuscript has been revised according to the reviewers' comments. All concerns raised by the reviewers have been adequately addressed by the authors and detailed responses to these comments are provided.